# Venture Capital, Compensation Incentive, and Corporate Sustainable Development

## Li Jing * and Huying Zhang

College of Management and Economics, Tianjin University, Tianjin 300072, China; hyzhang@tju.edu.cn
* Correspondence: jingli@tju.edu.cn; Tel.: +86-185-3840-5266

**Abstract:** Innovation is one of the primary approaches by which companies address the progressively severe social, environmental, and market pressures that they face, and it is a crucial route for companies to maintain sustainable development. Venture capital (VC) plays a significant role in promoting enterprise innovation, especially breakthrough innovation. Venture capital can increase executive compensation and corporate innovation. Previous studies have also indicated that compensation incentives can be beneficial to corporate innovation. Although the relationships between two of these three variables have been validated, the relationship between VC, executive compensation, and corporate innovation has not yet received ample consideration. Our research focuses on the connections among these three variables, and we chose corporate for our sample, which listed corporations on the Shenzhen and Shanghai stock exchanges in the period from 2009 to 2017. We found that VC has a mediating effect on innovation through executive compensation incentives, although not necessarily a full mediation effect—merely a partial one. Moreover, we found that VC primarily plays the role of a compensation incentive by amplifying the internal salary gap of corporate. By employing invention patents to replace explanatory variables, using a Heckman two-stage method, and utilizing propensity score matching (PSM) for robustness testing, the validity of the conclusion was confirmed. In addition, we discovered that experienced VC or companies with lower governance quality are more likely to use compensation incentives to promote corporate innovation. This study provides valuable insight for VC in cultivating corporate innovation, as well as for corporates looking to boost their innovation.

**Keywords:** venture capital; compensation incentive; corporate innovation; compensation gap

## 1. Introduction

Faced with increasingly intensified social, environmental, and market competition pressures, innovation is one of the primary means by which companies can achieve sustainable growth [1,2]. Firstly, through innovation in production processes, methods, and products, companies can not only save resource costs, reduce environmental pollution, and alleviate environmental pressures but also develop more environmentally friendly and socially responsible products and services, which further enhance their competitive advantage and social reputation in the market [3,4]. Secondly, innovation can help companies to optimize internal processes, reduce costs, and develop new products, which can improve production efficiency, lower production costs, and realize higher profits, thereby maintaining market competitiveness [5].

VC plays a significant role in promoting enterprise innovation, especially breakthrough innovation [6,7]. Studies have demonstrated that VC can provide value-added services to stimulate corporate innovation, such as offering funds, managerial expertise, and certifications. For instance, VC can decrease businesses' reliance on debt by providing them with funds to directly supplement research and development funds, thus advancing corporate innovation [6,8,9]. Additionally, VC can also furnish professional human resources, form alliances with other businesses, and enhance corporate governance [10–13].

Moreover, VC can reduce information asymmetry between corporate and external investors and increase corporate access to external resources through providing certifications [14,15]. This can help raise the chances of corporate obtaining loans [4] while decreasing financial constraints [16,17].

Studies have shown that VC can directly influence executive compensation, or indirectly impact executive salaries through engagement in corporate governance and raising the wages of the corporate employees that they invest in [6,18,19]. Studies have revealed that corporate executives backed by VC hold almost twice as many stock options compared with their non-supported counterparts [6], and employees' salaries are 10% higher than those of traditional competitors [20].

It is also suggested that VC investments have the potential to boost innovation through increased executive compensation. However, although this may arguably be the case, whether VC investments actually lead to innovations driven by increased remuneration has yet to be determined. Due to VC's short-term nature, its incentives for short-term performance can lead to unintended effects on corporate innovation. For example, increased incentive compensation may cause executives to prioritize quickly taking the company public, leading to underpricing in their initial public offering [21]. In the exit cycle, the focus on maximizing profits might encourage executives to increase share prices, which could then result in high executive bonuses [18]. Furthermore, young venture capitalists seeking to establish a successful track record are more likely to raise executive compensation to incentivize IPOs, mergers, and acquisitions [18,21].

We attempt to test the relationship between VC, incentive compensation, and corporate innovation. By using corporations listed on the Shenzhen and Shanghai stock exchanges from 2009 to 2017 as samples, we found that a link exists between VC's indirect promoting effect on corporate innovation and an increase in executive compensation; however, this is only a partial mediating effect. Furthermore, we also found that VC can increase the internal pay gap of the enterprise, which is consistent with the tournament theory, namely that VC can encourage employees to work harder, improving the corporation's innovation performance by increasing the internal pay gap. In addition, we also found that only high-experience venture capitalists or low-governance-quality firms adopted the strategy of increasing executive compensation.

Our research contributes to two aspects. Firstly, we verify the mechanism by which VC promotes corporate innovation through improving executive compensation. Our study shows that the mediation effect of VC in promoting corporate innovation by increasing executive compensation does exist, albeit with only a partial mediation effect. This provides a new interpretation for an increased understanding of the influence mechanism of VC on corporate innovation. In addition, following Yi, Rui [6], Shuwaikh [22], and Shin [23], our research supplements the literature on VC and corporate innovation.

Secondly, following Yi, Rui [6], Sun [18], and Kim [20], we determined that executives are rewarded with higher salaries than other employees for their ability to find investment opportunities. The salary gap increases employees' enthusiasm and reduces executives' "opportunity" behaviors, thus improving performance. Our study confirms that venture capital promotes enterprise innovation through enlarging the salary gap within companies. This not only verifies the tournament theory but also provides new evidence from the VC perspective.

The structure of the following study is as follows: a literature review and proposition of our hypotheses, our research design, an empirical test, and a discussion and conclusion.

## 2. Theoretical Analysis and Hypotheses

### 2.1. VC on Corporate Innovation

VC is beneficial in improving corporate innovation [6,7,12,13,24]. Previous studies have indicated that VC investment primarily increases innovation by providing capital, resources, and certifications. Firstly, VC investments provide access to funding, which can reduce debt dependency and directly help R&D activities [6,8,9]. This gives firms the

opportunity to build up reserves of research and development funds, ultimately increasing innovation [23].

Secondly, VC investments are able to offer a variety of resources to corporate, improving their innovation. Quas et al. found that VC contribution to total corporate asset growth was 38.13% [11]. VC may provide services to companies such as human resources and professional consultancy services, enabling them to actively recruit board directors and managers [10–13]. VC can also help build connections between firms through establishing alliances, especially when corporate employees receive the same VC investments [25–27]. Johnson et al. discovered that VC could provide more industry connections to corporate employees [28]. González et al. found an increased chance of resource exchange between corporate employees concerning innovation when both corporations receive VC investments from the same source, leading to improved efficiency of innovation [10].

Thirdly, VC may reduce information asymmetry between firms and external investors through certificating activities, potentially increasing access to external sources of financing [16], and reducing corporate financial constraints [17].

Finally, VC increases the level of corporate governance and reduces the costs associated with executive agency, leading to increased operational performance. VC monitoring has been shown to reduce executive agency costs, encourage risk-taking behavior, and promote innovation [29–31]. Additionally, VC monitoring improves internal corporate governance quality [32]. Hochberg et al. discovered that firms with VC had comparably low levels of earnings management during the IPO phase [30].

However, some scholars believe that the relationship between VC and corporate innovation is mainly due to the fact that VC can help corporate to select innovative employees [33]. Research by Bottazzi et al. found that European VC does not promote corporate innovation nor provide services, such as personnel recruitment [34]. Park et al. found that, in the later stages of investment, VC had a negative effect on corporate innovation [35]. In addition, some studies found that young venture capitalists are eager to push corporations into the market to quickly establish their reputation, resulting in higher discounts for corporate IPOs and reducing corporate innovation [21,36].

### 2.2. VC and Compensation Incentive

VC can directly or indirectly promote executive compensation for invested corporate employees, either by participating in the executive compensation contract formulation or by participating in corporate governance, respectively [6,18,37]. Almost twice as many executive stock options are acquired by VC-supported corporate employees compared with corporate employees without VC support [20], and their compensation is also 10% higher [38].

Before the corporate initial public offering (IPO), VC, eager to promote it, often guides executives through remuneration incentives to take the corporation public [39]. At the later stages of investment, VC actively increases executive remuneration to attain the maximum return by encouraging executives to drive up stock prices [40]. Additionally, young venture capitalists are eager to quickly establish their reputation, often increasing executive remuneration to incentivize taking companies public, and to encourage mergers, acquisitions, etc. [40,41]. Yi Cui found that among VC portfolios, due to the "contagion effect" of excess returns, executives were more likely to receive excess returns [6].

Additionally, venture capitalists that actively engage in corporate governance can issue compensation based on executive action information, thus reducing the sensitivity between executive pay and corporate performance [42,43]. Chen et al., Cao, Engel, and others have found that VC can significantly decrease the sensitivity between executive compensation and corporate performance [42]. However, Sun et al. observed that, in the later stages of investment, venture capitalists seeking higher returns will increase the sensitivity between executive compensation and stock return rates [18]. Lu et al. additionally determined that corporate employees supported by private equity funds had a heightened sensitivity between executive compensation and corporate performance [44].

### 2.3. Compensation Incentives and Corporate Innovation

Due to the "offset nature" of corporate performance, especially innovation, which requires multiple and larger investments and has greater uncertainty, the economic value brought by innovation can take a long time to manifest, and executives and owners often have conflicts of interest between long- and short-term benefits. In addition, in the standard agency contract, corporate executives are unable to obtain matching compensation for taking innovation risks to achieve positive outcomes. Therefore, executives tend to invest resources in low-risk projects and have a "natural" aversion to high-risk projects. To alleviate the conflicts of interest between executives and corporate owners, higher salaries (monetary and stock-option compensation) can be provided to the former, thus compensating for the cost of "bearing the risks of innovation failure" and realizing the "bundling" of executive returns and shareholder interests. Studies have shown that compensating managers for their risk-taking efforts can encourage corporate innovation [45–48]. Baranchuk et al. found that executives were more motivated to pursue innovation when given longer unexercised options [49]. Manso et al. also studied the incentives for optimal innovation and found that the best incentive schemes could be achieved through long-term stock-option grants [50].

Excessive compensation can also generate a substantial fear gap among employees. Previous studies have suggested that excessively high levels of compensation can cause managers to become more risk-averse, leading to decreased innovation performance in corporate settings [19,51]. Sun studied the effects of VC on executive compensation and found that its impact on executive compensation is more consistent with grandstanding in the exit stage [18]. Chahine found that young VC option grants bribe CEOs to agree to an early IPO [21].

Previous literature has examined the relationship between any two of the three factors of VC, compensation incentives, and corporate innovation; however, there has been no unified study on the relationship between all three. Although it seems logical that VC can be leveraged to enhance corporate innovation through compensation incentives, there has been no relevant empirical validation.

### 2.4. Hypotheses

Given the high-risk and long-term nature of innovation, executives, as agents, are naturally risk-averse under the pressures of performance appraisals [45,52]. Additionally, under fixed-wage contracts, executives bear the risks of unsuccessful innovation, while being unable to share in the rewards of successful innovations. By providing compensation incentives, not only can executive gains be linked to shareholder benefits [53], the incentives can also compensate for the cost of taking risks in innovation, thus promoting corporate innovation [47,48,52]. Previous studies have suggested that VC investments can increase executive compensation [54,55].

Compared with the lock-up requirements for listed companies' stocks in other countries, the securities regulatory authorities in China require the shareholders of listed companies to have a longer lock-up period (often three years or even longer). Consequently, risk investors are not able to immediately exit as in other markets; instead, they are required to hold their shares for a much longer period. To achieve higher returns from investments, VC has an incentive to encourage executives to invest in innovative resources and promote sustainable corporate growth.

Moreover, in the VC market, high-reputation venture capitalists are not only able to finance more easily but are also able to obtain more bargaining power in the investment process. According to previous studies, if the VC-backed corporations have a large drop in performance level, the reputation of the VC would suffer, which would be detrimental to subsequent financing and investing processes. In conclusion, VC has a motive to provide executives with incentives to "compensate" them for the risks taken in the innovation process, prompting them to actively invest in innovative resources, promoting corporate innovation. In conclusion, Hypothesis 1A is proposed:

**Hypothesis 1A:** *VC can promote corporate innovation by increasing executive compensation.*

VC may increase executive compensation to push forward a company's IPO or boost stock prices for exits. Compensation incentives might cause executives to be eager to take companies public with a relatively lower price [18]. During the later phase of investment, VC is likely to actively increase executive compensation to maximize returns [9], and young VC also tends to increase executive salaries to build reputation quickly, incentivizing them to take their corporation public through IPOs, mergers, acquisitions, and so on [18,21]. Sun et al., however, found that during the later investment phase, VC may increase the sensitivity between executive compensation and stock return rates to achieve higher returns, which can restrain corporate innovation [18]. In conclusion, the short-term VC performance seeking may harm a firms' long-term performance; therefore, Hypothesis 1B is proposed:

**Hypothesis 1B:** *Venture capital can increase executive compensation levels but does not improve corporate innovation.*

## 3. Research Design

### 3.1. Sample Selection and Data Source

We chose corporations listed on the Shenzhen and Shanghai stock exchanges in the period from 2009 to 2017 as our sample. To exclude interference factors, first, due to the focus on the innovation ability of the enterprises, we chose manufacturing enterprises with relatively strong innovation capabilities. Second, we excluded enterprises with zero cumulative patent applications before the IPO. Finally, we obtained 868 samples.

To identify whether the corporation received VC, we first checked if the names of the top ten shareholders included equity investment or venture investment, if so, the corporation received VC; otherwise, we proceeded to the next step. Then, we further matched the corporation's name with the Qingke database. If the corporation was found in the Qingke database, it received venture capital investments; otherwise, it did not receive anything. Qingke is a relatively authoritative database in China that records venture capital investments. Other databases used in the sample came from CSMAR.

### 3.2. Econometric Modeling

Following the mediation test proposed by Baron [56] and Imai [57], we utilized the following equation to assess the mediating effect of VC on corporate innovation through compensation incentives:

$$\text{Lnpat} = \text{vcif} + \text{control} + \text{ind} + \text{year} \tag{1}$$

$$\text{ms} = \text{vcif} + \text{control} + \text{ind} + \text{year} \tag{2}$$

$$\text{Lnpat} = \text{vcif} + \text{ms} + \text{control} + \text{ind} + \text{year} \tag{3}$$

where lnpat represents firm innovation output, vcif denotes whether firms have received VC, ms indicates managerial compensation level, and control encompasses variables such as firm age (age), size, return on assets (roa), cash, leverage (lev), R&D investment (rd), operational efficiency (oer), shareholder concentration (shard), board size (bdn), and fraction of independent directors (ibd). Year and ind denote year and industry, respectively.

### 3.3. Measurement of Variables

The dependent variable: innovation. This paper selects the number of patent applications by corporate as an indicator for innovation. In the literature on corporate innovation, it is common for scholars to use patent applications, patents obtained, and total factor productivity as indicators for measuring innovation [58–61]. Although using patent indicators to measure corporate innovation is very coarse, considering that patent data are relatively

objective, many studies typically use the number of patents to measure corporate innovation [62,63]. Moreover, in view of the volatility of R&D innovation output of corporate, this paper takes the cumulative number of patent applications by corporate in the three years before IPO as an indicator to measure innovation. Relevant data are sourced from the innojoy database. The logarithm of the number of patents is denoted as lnpat.

Independent variable. VC. Following the definition of VC variables given by Sun and Liao [18,32], this paper uses 1 to denote receiving VC, while 0 implies no such investment.

Executive compensation. Following the definition of executive compensation given by Hen, Conyon, Cao, and Firth [64–66], this paper uses remuneration received by directors, supervisors, and executives as incentives for executive compensation, which is denoted by ms.

Control variables. Referencing the previous relevant literature [67], corporate age, asset scale, asset return rate, asset liability ratio, operating efficiency, cash flow, research and development investment, equity concentration, size of the board of directors, proportion of independent directors, listing year, and industry are considered to be key factors affecting corporate performance; therefore, we selected these variables as control. Table 1 reports the description of variables, measures.

**Table 1.** Variable definition.

| Variable | Definition |
|---|---|
| Lnpat | The actual total number of application patents in the three years before IPO is converted into a logarithmic value. |
| ms | The total of the salaries of directors, supervisors, and senior executives. |
| age | The logarithm of the difference between the year of listing and the year of company's establishment. |
| size | The logarithm of the total assets of the corporation. |
| roa | Return on assets. |
| cash | Cash possessed by enterprises. |
| lev | Asset–liability ratio. |
| rd | Logarithm of R&D investment. |
| oer | Ratio of operating cost to operating income. |
| shard | Total shareholding ratio of the top three shareholders. |
| bdn | Board size. |
| ibd | Proportion of independent directors. |

## 4. Empirical Results and Analysis

### 4.1. Descriptive Statistics and Correlation Analysis

Table 2 shows the minimum value of lnpat is 0.693, the maximum value is 6.188, and the variance is 1.105, suggesting substantial differences in innovation levels among corporations. The minimum value of ms is 0.374, the mean is 2.26, and the maximum value is 18.624, implying large differences in executive compensation among samples. The correlation between lnpat and vcif is 0.108, which is significant at the 1% level. The correlation between lnpat and ms is 0.258 and is also significant at the 1% level. The correlation between vcif and ms is 0.106 and is significant at the 1% level. These initial results suggest that there is a relationship between vcif and both ms and lnpat.

**Table 2.** Descriptive statistics.

| Variable | Mean | Variance | lnpat | vcif | ms | age | size | roa | cash | lev | rd | oer | shard | bdn |
|---|---|---|---|---|---|---|---|---|---|---|---|---|---|---|
| lnpat | 3.39 | 1.11 | 1.00 | | | | | | | | | | | |
| vcif | 0.57 | 0.50 | 0.108 *** | 1.00 | | | | | | | | | | |
| ms | 2.26 | 1.67 | 0.258 *** | 0.106 *** | 1.00 | | | | | | | | | |
| age | 12.52 | 4.86 | 0.123 *** | 0.04 | 0.111 *** | 1.00 | | | | | | | | |
| size | 19.91 | 0.74 | 0.269 *** | 0.059 * | 0.378 *** | 0.176 *** | 1.00 | | | | | | | |
| roa | 0.10 | 0.05 | 0.02 | −0.105 *** | 0.03 | −0.03 | −0.260 *** | 1.00 | | | | | | |
| cash | 18.14 | 0.82 | 0.248 *** | 0.068 ** | 0.355 *** | 0.121 *** | 0.701 *** | −0.02 | 1.00 | | | | | |
| lev | 0.42 | 0.15 | 0.090 *** | −0.03 | 0.05 | −0.05 | 0.469 *** | −0.613 *** | 0.155 *** | 1.00 | | | | |
| rd | 8.52 | 0.80 | 0.332 *** | 0.075 ** | 0.469 *** | 0.162 *** | 0.670 *** | −0.04 | 0.581 *** | 0.186 *** | 1.00 | | | |
| oer | 0.83 | 0.10 | 0.114 *** | 0.03 | 0.01 | 0.112 *** | 0.336 *** | −0.688 *** | 0.097 *** | 0.573 *** | 0.229 *** | 1.00 | | |
| shard | 56.24 | 12.42 | 0.02 | −0.168 *** | −0.03 | −0.05 | 0.102 *** | 0.02 | 0.062 * | 0.02 | 0.01 | −0.03 | 1.00 | |
| bdn | 8.32 | 1.41 | 0.03 | 0.070 ** | 0.061 * | −0.03 | 0.171 *** | −0.120 *** | 0.147 *** | 0.092 *** | 0.114 *** | 0.098 *** | −0.160 *** | 1.00 |
| ibd | 0.37 | 0.05 | 0.04 | −0.141 *** | −0.01 | 0.058 * | −0.073 ** | 0.110 *** | −0.04 | −0.067 ** | −0.03 | −0.102 *** | 0.137 *** | −0.608 *** |

Note: * $p < 0.10$, ** $p < 0.05$, *** $p < 0.01$.

### 4.2. Regression Results and Analysis

Table 3 examines the mediating effect of VC in promoting corporate innovation. column 1 is the relationship between corporate characteristics and innovation, while column 2 tests the impact of VC on corporate innovation. column3 is the examination of the impact of VC on executive compensation, and column 4 is used to test the relationship between venture capital investments, executive compensation, and corporate innovation. Column 2–4 are the mediating effects of incentive compensation tests on the role of VC in promoting corporate innovation. Incolumn2, the coefficient of vcif is 0.141 and is significantly not equal to 0 at the 5% level of significance, indicating that VC has a promotional role in corporate innovation output. In column 3, the vcif coefficient is 0.240 and is significantly not equal to 0 at the 5% level of significance, indicating that VC has a promotional role in executive compensation. In column 4, the vcif coefficient is 0.115 and is significantly not equal to 0 at the 10% level of significance, and the MS coefficient is 0.107 and is significantly not equal to 0 at the 1% level of significance. Additionally, the vcif coefficient (0.115) in column 4 is lower than the coefficient (0.141) in column 2, suggesting that VC has a partial mediating effect on promoting enterprise innovation through increasing executive compensation. The magnitude of this mediating effect is 0.02568 ($0.107 \times 0.240$), with an explanatory power of 18.21% (0.02568/0.141). We use a Sobel test to further verify the mediating effect, and the Z value is 1.863, which is significantly not equal to 0 at the 10% level of significance. Consequently, Hypothesis 1A is accepted; that is, VC promotes corporate innovation by raising executive compensation.

**Table 3.** Test of the intermediary effect of VC promoting innovation.

|  | 1 | 2 | 3 | 4 |
|---|---|---|---|---|
|  | lnpat | lnpat | ms | lnpat |
| vcif |  | 0.141 ** | 0.240 ** | 0.115 * |
|  |  | (2.07) | (2.31) | (1.71) |
| ms |  |  |  | 0.107 *** |
|  |  |  |  | (4.73) |
| age | 0.012 | 0.012 | 0.005 | 0.012 |
|  | (1.58) | (1.64) | (0.48) | (1.58) |
| size | −0.039 | −0.035 | 0.328 ** | −0.070 |
|  | (−0.41) | (−0.36) | (2.24) | (−0.74) |
| roa | 1.989 ** | 2.345 ** | 0.249 | 2.319 ** |
|  | (1.96) | (2.29) | (0.16) | (2.29) |
| cash | 0.151 ** | 0.146 ** | 0.116 | 0.134 ** |
|  | (2.40) | (2.32) | (1.20) | (2.15) |
| lev | 0.043 | 0.098 | 0.141 | 0.083 |
|  | (0.12) | (0.28) | (0.26) | (0.24) |
| rd | 0.319 *** | 0.315 *** | 0.767 *** | 0.233 *** |
|  | (5.09) | (5.03) | (8.00) | (3.63) |
| oer | 0.931 * | 0.982 * | −2.324 *** | 1.231** |
|  | (1.78) | (1.88) | (−2.91) | (2.37) |
| shard | 0.000 | 0.000 | 0.005 | −0.000 |
|  | (0.00) | (0.03) | (0.66) | (−0.08) |
| bdn | 0.047 * | 0.042 * | −0.005 | 0.043 * |
|  | (1.86) | (1.67) | (−0.14) | (1.72) |
| _cons | −3.198 ** | −3.308 ** | −11.480 *** | −2.082 |
|  | (−2.35) | (−2.44) | (−5.52) | (−1.53) |
| ind | Yes | Yes | Yes | Yes |
| year | Yes | Yes | Yes | Yes |
| N | 868 | 868 | 868 | 868 |
| $R^2$ | 0.317 | 0.321 | 0.300 | 0.339 |
| $R^2$_adjust | 0.274 | 0.277 | 0.255 | 0.295 |

Note: * $p < 0.1$, ** $p < 0.05$, *** $p < 0.01$. Standard errors in parentheses.

## 5. Further Study

### 5.1. Sub-Sample Analysis

We have shown that VC can promote corporate innovation by raising executive salaries. However, due to high-experience VC being able to identify the motives of executives, other measures may be taken to increase corporate innovation, such as appointing directors, making frequent visits to corporate, and so on [68–70]. Therefore, VC of different experience levels may take different measures to promote corporate innovation. The sample was divided into two groups based on the VC experience level: a low- and high-experience group. Table 4 shows that while the coefficient of vcif in column 1 (low-experience group) was 0.476, which was not statistically significant, the coefficient of vcif in column 2 (high-experience group) was 0.337, and was statistically significant at the 1% level. This indicates that only high-experience VC took measures to raise executive salary levels, which is consistent with prior research, meaning that high-risk investment has a greater impact on corporate innovation.

**Table 4.** Compensation-incentive behavior of VC in different situations.

| | 1 | 2 | 3 | 4 |
|---|---|---|---|---|
| | Lower experience | Higher experience | Higher quality | Lower quality |
| vcif | 0.476 | 0.337 *** | 0.040 | 0.481 *** |
| | (0.77) | (2.60) | (0.29) | (2.96) |
| age | 0.004 | 0.012 | 0.019 | −0.011 |
| | (0.19) | (0.83) | (1.38) | (−0.58) |
| size | 0.780 *** | 0.044 | 0.286 | 0.479 ** |
| | (2.74) | (0.25) | (1.46) | (2.07) |
| roa | −3.717 | 2.118 | 0.607 | −0.574 |
| | (−1.07) | (1.17) | (0.27) | (−0.24) |
| cash1 | 0.093 | 0.189 * | 0.137 | −0.045 |
| | (0.48) | (1.67) | (1.09) | (−0.29) |
| lev | −2.032 ** | 1.484 ** | 0.329 | −0.365 |
| | (−2.05) | (2.24) | (0.46) | (−0.44) |
| rd | 0.754 *** | 0.770 *** | 0.638 *** | 0.956 *** |
| | (3.93) | (6.64) | (5.31) | (5.92) |
| oer | −3.450 ** | −2.269 ** | −2.674 ** | −1.947 |
| | (−2.00) | (−2.45) | (−2.54) | (−1.51) |
| shard10 | 0.011 | 0.002 | 0.004 | 0.010 |
| | (0.73) | (0.23) | (0.51) | (0.87) |
| bdn | −0.044 | 0.047 | 0.070 | −0.053 |
| | (−0.60) | (1.01) | (1.29) | (−0.91) |
| _cons | −19.511 *** | −8.087 *** | −10.259 *** | −13.567 *** |
| | (−4.39) | (−3.31) | (−3.63) | (−4.23) |
| ind | Yes | Yes | Yes | Yes |
| year | Yes | Yes | Yes | Yes |
| N | 289 | 579 | 431 | 437 |
| $R^2$ | 0.354 | 0.333 | 0.330 | 0.349 |
| $R^2$_adjust | 0.241 | 0.267 | 0.246 | 0.270 |

Note: * $p < 0.1$,** $p < 0.05$, *** $p < 0.01$. Standard errors in parentheses.

VC adopts compensation incentives to promote corporate innovation due to managers' "natural aversion" to innovative behaviors and relatively lower corporate governance quality. In corporations with higher governance quality, managers' "agency problems" have been effectively solved, thus eliminating the issue of pursuing short-term, at the expense of long-term, performance. In corporations with better corporate governance qualities, VC monitoring may not be effective in reducing the agency costs of managers, which are well controlled. Thus, its impact would be larger in corporations with poorer corporate governance qualities. In this paper, we further reference Bai [71] and Jiang [59],

who built a corporate governance quality index using eight indicators (top shareholder holdings ratio, second–tenth shareholder holdings ratio, external director ratio, audit agency, whether listed cross-wise, whether it is a state-owned corporation, whether its CEO and chairman are the same individual, and whether there is a parent company). Column 3 and column 4 in Table 4 test the influence of VC on executive compensation under different corporate governance qualities for corporations with high- or low-governance qualities, respectively. The empirical results show that the vcif coefficient for column 3 is 0.040 but not significant, while in column 4, the coefficient reaches 0.481 and is significantly not 0 at the 1% significance level. Thus, it is only when associated with poor corporate governance quality that VC implements "indirect monitoring" to enhance executive compensation.

*5.2. VC, Compensation Gap, and Corporate Innovation*

Executives can create value for the corporation through participating in production decisions, and they possess the skills to discover new investment opportunities; thus, they are rewarded with higher compensation than other employees [72]. According to tournament theory, a larger salary gap can bring greater benefits to winners and also incentivize laggards to make more effort [73], which can enhance employees' enthusiasm, reduce executives' "opportunity" behavior [15], and improve corporate performance [74] and innovation output [75]. Therefore, VC is incentivized to increase the internal salary gap, resulting in increases in executive effort, to realize the goal of obtaining maximum profit. In this paper, we adopt the ratio of average executive compensation to average employee salary to measure the salary gap, following Faleye and Banker et al. [74,76]. Specifically, average executive salary is calculated as the total annual salaries of directors, supervisors, and executives divided by the managerial scale, which is the sum of "number of directors", "number of executives", and "number of supervisors", minus the "number of independent directors" and "number of directors, supervisors or executives not receiving salaries". Average employee salary is the change in the "total wages payable to employees" plus the "cash paid to or on behalf of employees" minus "the total annual salaries of directors, supervisors and executives" divided by the number of employees. The main explanatory variable "corporate salary gap" is obtained by calculating the ratio mentioned above, which is denoted by Gap.

Table 5 shows the effects of VC on the executive compensation gap, in which Column 1 is the relationship between VC and corporate innovation; Column 2 is the relationship between compensation gap and corporate innovation; Column 3 tests the relationship between venture capital investment and compensation gap; and column 4 tests the relationship among VC, compensation gap, and corporate innovation. Table 5 shows that the vcif coefficient in column 1 is 0.238 and is significantly not 0 at the 1% significance level, indicating the positive promoting effect of VC on corporate innovation. The gap coefficient in column 2 is 0.120, which is also significantly not 0 at the 1% significance level, implying a positive promoting effect of compensation gap on corporate innovation. The vcif coefficient in column 3 is 0.173 and is significantly not 0 at the 5% significance level, suggesting a positive promoting effect of VC on compensation gap. In column 4, the vcif coefficient is 0.173 and the gap coefficient is 0.115, and they are both significantly not 0 at the 5% and 1% significance levels. Specifically, the vcif coefficient in column 4 (0.173) is smaller than that in column 1 (0.238), implying that VC promotes corporate innovation through increasing the executive compensation gap; however, this mediating effect is partial, with an influence size of 0.05704 (0.115 × 0.496) and an explanatory power of 23.97% (0.05704/0.238). To further test the mediating effect, we used the Sobel test, yielding a Z value of 2.772, which is significantly not 0 at the 1% significance level. Consequently, VC promotes corporate innovation by increasing the internal compensation gap.

**Table 5.** Testing the relationship between VC, compensation gap, and corporate innovation.

| | 1 | 2 | 3 | 4 |
|---|---|---|---|---|
| | lnipat | lnipat | gap | lnipat |
| vcif | 0.238 *** | | 0.496 *** | 0.173 ** |
| | (3.04) | | (3.38) | (2.24) |
| gap | | 0.120 *** | | 0.115 *** |
| | | (6.48) | | (6.18) |
| age | 0.008 | 0.010 | −0.003 | 0.009 |
| | (0.79) | (1.06) | (−0.19) | (0.90) |
| size | −0.119 | −0.258 ** | 1.180 *** | −0.246 ** |
| | (−1.09) | (−2.37) | (5.80) | (−2.26) |
| roa | 0.463 | 0.887 | −0.781 | 0.656 |
| | (0.23) | (0.44) | (−0.20) | (0.33) |
| cash | 0.156 ** | 0.161 ** | −0.035 | 0.162 ** |
| | (2.15) | (2.26) | (−0.26) | (2.28) |
| lev | −0.313 | −0.124 | −1.222 | −0.162 |
| | (−0.67) | (−0.27) | (−1.41) | (−0.36) |
| rd | 0.520 *** | 0.462 *** | 0.491 *** | 0.456 *** |
| | (7.15) | (6.41) | (3.60) | (6.34) |
| oer | 0.099 | 0.349 | −1.176 | 0.237 |
| | (0.13) | (0.47) | (−0.83) | (0.32) |
| shard | −0.004 | −0.005 | 0.013 | −0.006 |
| | (−0.72) | (−0.87) | (1.24) | (−1.01) |
| bdn | 0.049 | 0.048 | −0.095 | 0.058 |
| | (1.02) | (1.03) | (−1.07) | (1.23) |
| IMR | 0.054 | −0.126 | 0.187 | 0.013 |
| | (0.15) | (−0.37) | (0.29) | (0.04) |
| _cons | −2.981 * | −0.324 | −21.937 *** | −0.585 |
| | (−1.93) | (−0.21) | (−7.58) | (−0.37) |
| ind | Yes | Yes | Yes | Yes |
| year | Yes | Yes | Yes | Yes |
| N | 835 | 832 | 832 | 832 |
| $R^2$ | 0.244 | 0.273 | 0.254 | 0.277 |
| $R^2$_adjust | 0.205 | 0.235 | 0.215 | 0.239 |

Note: * $p < 0.1$, ** $p < 0.05$, *** $p < 0.01$. Standard errors in parentheses.

## 6. Robustness Test

We mainly checked the robustness of our conclusion by replacing explanatory variables (using invention patents) and using Heckman two-stage estimation and propensity score matching (PSM) tests.

First, we used invention patents instead of all patents as dependent variables because invention patents are better when reflecting an innovative ability; therefore, we took their natural logarithm and denoted it as lnipat [62]. The coefficients of vcif in column 1–3 in Table 6 are 0.213 (1% level of significance), 0.240 (5% level of significance), and 0.184 (5% level of significance), respectively. These coefficients indicate that VC can promote corporate innovation and increase executive compensation, with partial mediation involved in the positive effect of VC on corporate innovation, and the size and explanatory capacity being 0.02568 (0.107 × 0.240) and 18.21% (0.02568/0.141), respectively. For further validation, a Sobel test is performed, yielding a Z value of 1.863, which is significantly different from 0 at the 10% significance level. Thus, Hypothesis 1A is supported, showing that VC can enhance corporate innovation by increasing executive compensation, albeit with partial mediation.

**Table 6.** The robustness test of the mediating effect of VC on corporate innovation.

|  | 1 | 2 | 3 |
|---|---|---|---|
|  | lnipat | ms | lnipat |
| vcif | 0.213 *** | 0.240 ** | 0.184 ** |
|  | (2.81) | (2.31) | (2.45) |
| ms |  |  | 0.123 *** |
|  |  |  | (4.91) |
| age | 0.009 | 0.005 | 0.008 |
|  | (1.08) | (0.48) | (1.01) |
| size | −0.100 | 0.328 ** | −0.140 |
|  | (−0.94) | (2.24) | (−1.33) |
| roa | 0.449 | 0.249 | 0.418 |
|  | (0.39) | (0.16) | (0.37) |
| cash1 | 0.139 ** | 0.116 | 0.125 * |
|  | (1.99) | (1.20) | (1.81) |
| lev | −0.331 | 0.141 | −0.348 |
|  | (−0.84) | (0.26) | (−0.90) |
| rd | 0.523 *** | 0.767 *** | 0.429 *** |
|  | (7.50) | (8.00) | (6.00) |
| oer | 0.064 | −2.324 *** | 0.351 |
|  | (0.11) | (−2.91) | (0.61) |
| shard10 | −0.003 | 0.005 | −0.004 |
|  | (−0.67) | (0.66) | (−0.79) |
| bdn | 0.048 * | −0.005 | 0.049 * |
|  | (1.71) | (−0.14) | (1.76) |
| _cons | −3.062 ** | −11.480 *** | −1.645 |
|  | (−2.02) | (−5.52) | (−1.08) |
| ind | Yes | Yes | Yes |
| year | Yes | Yes | Yes |
| N | 868 | 868 | 868 |
| $R^2$ | 0.263 | 0.300 | 0.284 |
| $R^2$_adjust | 0.215 | 0.255 | 0.236 |

Note: * $p < 0.1$, ** $p < 0.05$, *** $p < 0.01$. Standard errors in parentheses.

Second, VC may exhibit a "screening" effect, meaning that VC may invest in corporations with stronger innovation capabilities. In this paper, we utilize Heckman's two-stage method to examine the mediating effect of VC when promoting corporate innovation [77,78], whereby in the first stage the corporate features are used to predict whether VC invests and calculates the inverse Miller factor (IMR). In the second stage, IMR is entered as a variable into the equation. Table 7 shows that by using the probit method in column 1 to predict whether the corporation receives VC, the IMR is obtained. For column 2, the vcif coefficient is 0.140 and is significantly different from 0 at the 5% significance level. Furthermore, for column 3, the vcif coefficient is 0.239 and is significantly different from 0 at the 5% significance level, indicating that VC has a supportive role on executive salary. In column 4, the vcif coefficient is 0.115 and is significantly different from 0 at the 10% significance level, and its coefficient is lower than that of column 2, implying that VC promotes corporate innovation through increasing executive salary; however, it is only partially raised as a mediator, with the size of its impact being 0.02557 ($0.107 \times 0.239$) and its explanatory power being 18.27% (0.02568/0.140). To further test the mediating effect, in this paper, we used the Sobel test, obtaining a Z value of 1.844, which is significant at the 10% significance level. Even after controlling for sample self-selection, our conclusion still stands.

**Table 7.** The Heckman two-stage test to examine the mediating effect of VC promoting corporate innovation.

|  | **1** | **2** | **3** | **4** |
|---|---|---|---|---|
|  | vcif | lnpat | ms1 | lnpat |
| vcif |  | 0.140 ** | 0.239 ** | 0.115 * |
|  |  | (2.06) | (2.29) | (1.70) |
| ms1 |  |  |  | 0.107 *** |
|  |  |  |  | (4.71) |
| age | −0.008 | 0.009 | −0.001 | 0.009 |
|  | (−0.77) | (0.91) | (−0.06) | (0.94) |
| size | −0.095 | −0.091 | 0.229 | −0.116 |
|  | (−0.83) | (−0.69) | (1.13) | (−0.88) |
| roa | −6.857 *** | −1.545 | −6.551 | −0.848 |
|  | (−4.81) | (−0.24) | (−0.67) | (−0.13) |
| cash | 0.080 | 0.188 ** | 0.189 | 0.168 * |
|  | (0.99) | (2.03) | (1.33) | (1.83) |
| lev | −0.871 * | −0.360 | −0.659 | −0.289 |
|  | (−1.88) | (−0.44) | (−0.52) | (−0.36) |
| rd | 0.098 | 0.369 *** | 0.862 *** | 0.277 ** |
|  | (1.24) | (3.41) | (5.20) | (2.55) |
| oer | −1.085 * | 0.348 | −3.432 * | 0.714 |
|  | (−1.68) | (0.30) | (−1.94) | (0.63) |
| shard | −0.003 | −0.001 | 0.002 | −0.001 |
|  | (−0.44) | (−0.25) | (0.28) | (−0.30) |
| bdn | 0.076 ** | 0.083 | 0.066 | 0.076 |
|  | (2.29) | (1.17) | (0.61) | (1.08) |
| IMR |  | 0.906 | 1.583 | 0.737 |
|  |  | (0.62) | (0.70) | (0.51) |
| _cons | 1.057 | −3.317 ** | −11.496 *** | −2.093 |
|  | (0.67) | (−2.44) | (−5.53) | (−1.53) |
| ind | No | Yes | Yes | Yes |
| year | Yes | Yes | Yes | Yes |
| N | 868 | 868 | 868 | 868 |
| $R^2$ |  | 0.321 | 0.301 | 0.339 |
| $R^2$_adjust |  | 0.276 | 0.254 | 0.294 |

Note: * $p < 0.1$, ** $p < 0.05$, *** $p < 0.01$. Standard errors in parentheses.

Finally, in this paper, we adopt propensity score matching (PSM) and divide corporations into experimental and control groups [33]. In Table 8, column 1 shows the impact of vcif on corporate innovation, with a coefficient of 0.141 and statistical significance at the 5% level. Column 4 reveals the influence of VCIF on the salary level of senior executives, with a coefficient of 0.240, which is significant at the 5% level. Column 4 is the effect of both vcif and ms on corporate innovation; vcif has a coefficient of 0.115 and is statistically significant at the 10% level, and ms has a coefficient of 0.107 and is significant at the 1% level. Compared with column 1, column 3 shows a lower coefficient for vcif, indicating that there is a mediating effect of VC promoting corporate innovation by increasing the salary of executives; however, it is only a partial mediation, with an influence of 0.02568 (0.107 × 0.240) and an explaining ability of 18.21% (0.02568/0.141). Furthermore, we used the Sobel test to further examine the mediation effect, yielding a Z value of 1.863, which is significant at the 10% significance level.

**Table 8.** Using the PSM to examine the mediating effect of VC promoting corporate innovation.

| | **1** | **2** | **3** |
|---|---|---|---|
| | lnpat | Ms | Lnpat |
| vcif | 0.141 ** | 0.240 ** | 0.115 * |
| | (2.07) | (2.31) | (1.71) |
| ms | | | 0.107 *** |
| | | | (4.73) |
| | (1.64) | (0.48) | (1.58) |
| size | −0.035 | 0.328 ** | −0.070 |
| | (−0.36) | (2.24) | (−0.74) |
| roa | 2.345 ** | 0.249 | 2.319 ** |
| | (2.29) | (0.16) | (2.29) |
| cash1 | 0.146 ** | 0.116 | 0.134 ** |
| | (2.32) | (1.20) | (2.15) |
| lev | 0.098 | 0.141 | 0.083 |
| | (0.28) | (0.26) | (0.24) |
| rd | 0.315 *** | 0.767 *** | 0.233 *** |
| | (5.03) | (8.00) | (3.63) |
| oer | 0.982 * | −2.324 *** | 1.231 ** |
| | (1.88) | (−2.91) | (2.37) |
| shard10 | 0.000 | 0.005 | −0.000 |
| | (0.03) | (0.66) | (−0.08) |
| bdn | 0.042 * | −0.005 | 0.043 * |
| | (1.67) | (−0.14) | (1.72) |
| _cons | −3.308 ** | −11.480 *** | −2.082 |
| | (−2.44) | (−5.52) | (−1.53) |
| ind | Yes | Yes | Yes |
| year | Yes | Yes | Yes |
| N | 868 | 868 | 868 |
| $R^2$ | 0.321 | 0.300 | 0.339 |
| $R^2$_a | 0.277 | 0.255 | 0.295 |

Note: * $p < 0.1$, ** $p < 0.05$, *** $p < 0.01$. Standard errors in parentheses.

## 7. Discussion

### 7.1. Conclusions and Discussion

This paper mainly investigates the relationships among VC, compensation incentives, and corporate innovation. Our findings indicate that VC can promote corporate innovation by raising executive compensation; however, the salary incentive was only a partial mediator. Additionally, we found that VC promotes corporate innovation performance through enlarging the internal compensation gap, which encourages executives to work harder and is consistent with the tournament theory. Moreover, we further studied the particular contexts under which VC increases executive compensation performance. We discovered that high-quality VC or lower governance quality are more likely to embrace compensation-incentive measures. Robustness tests show that the conclusion remains valid.

Firstly, VC can promote corporate innovation through compensation incentives. VC can raise the level of executive salary, which may be because of the long-term performance of a corporation or the pursuit of short-term results. VC also has a positive impact on corporate innovation. However, the relationship between these three does not infer that VC prompts corporate innovation through raising executive salaries. Through verifying the relationship among these three factors, our research affirmed the mediated effect of VC on corporate innovation; however, it is a partially mediated effect. This implies that the impact of VC on corporate innovation is not solely reliant on increasing executive salaries; therefore, further research needs to be performed. We found that VC mainly realizes the compensatory effect by expanding the internal salary gap in the corporation, unveiling the concrete measures of VC. However, the expansion of the salary gap may lead to excessive

executive gains. This finding is consistent with conclusions from previous studies, namely that executives in VC-supported corporations often gain excessive returns [79,80]. Against the backdrop of the COVID-19 pandemic, the impact of corporate governance and social responsibility regarding sustainable development has been widely scrutinized [81–83]. Our findings suggest that venture capital can improve the innovation performance of enterprises by participating in corporate governance through enhancing managerial compensation, thereby affecting the sustainable development of enterprises. However, given the crucial importance of corporate social responsibility for sustainable development, the impact of venture capital on corporate social responsibility and sustainable development deserves further attention as a potential area of future investigation.

Secondly, VC is more likely to produce incentives in high-reputational or low-governance-quality corporations. By splitting up the sample, our research showed that executives are more likely to obtain incentives from high-reputation VC or low-governance-quality corporations. Prior studies have shown that high-reputation VC has a greater influence on corporate innovation, suggesting that high-reputation VC can provide more resources or be more engaged in corporate governance. Our research re-validated this point of view, providing new evidence that high-reputation VC can promote corporate innovation through compensation incentives. Furthermore, in lower-quality corporations, we found that executives are more likely to obtain higher salaries. This is mainly because in low-governance-quality corporations, VC does not have strong corporate monitoring; instead, it uses compensation incentives to guide executive input on innovation. This conforms to the conclusions from previous studies that corporations of different governance qualities have variations in the governance measures adopted by VC.

### 7.2. Theoretical Contribution

Our research contributes two new findings concerning the relationship between VC, compensation incentives, and corporate innovation. Firstly, our research provides a new explanation mechanism for how VC promotes innovation in corporations. Our findings not only provide new evidence for the promotion of corporate innovation by VC but also make up for the absence of an integrated framework for studying the linkage among all three variables in the previous literature. Furthermore, we examined the circumstances under which venture capital implements measures to raise executive salaries. Our results demonstrate how VC facilitates corporate innovation through increased executive compensation, which is more precisely differentiated than the approaches taken in earlier studies that only took the increase of executive salaries by VC into consideration.

Secondly, we incorporated internal salary disparity into our research framework, thereby elucidating the primary way in which VC boosts corporate innovation through broadened compensation gaps. Unlike the prior literature on compensation competition, we present proof of external shareholders wielding compensation incentives to administer a company.

### 7.3. Practical Implications

Our study has practical implications. To begin with, VC can increase a corporation's innovativeness, producing high returns through enhancing internal compensation. Our research demonstrates that VC can advance corporate innovation by augmenting managerial compensation, typically by augmenting the internal compensation gap. This provides a viable approach for VC to nurture corporations. In addition, high-reputation VC or corporations with low governance quality often opt to raise managerial compensation levels. Therefore, low-reputation VC can learn from high-reputation VC, facilitating corporate innovation by increasing executive compensation.

Moreover, from the viewpoint of corporate, corporations should increase executive compensation to motivate executives to invest in innovation resources and focus on long-term company development by compensating the risks that executives endure due to innovation. Furthermore, it is feasible to increase the internal compensation gap because the

value created by executives is greater than that created by other personnel in corporations, thus incentivizing executives to fulfill their responsibilities for higher compensation.

### 7.4. Limitations and Future Research

Despite our research confirming the mediating role of VC in promoting corporate innovation through increased executive compensation and utilizing various methods for robustness tests, some inadequacies remain. Firstly, while the previous literature usually utilized patent applications or grant rates, R&D expenditure, and total factor productivity, it should be noted that innovation involves a combination of corporate secrets or process and technology innovations, which cannot be accurately measured using patent data alone. Hence, more suitable indicators and reasonable metrics can be employed in the future for assessing corporate innovation. Secondly, this study substantiated the mediating role of VC in promoting corporate innovation via increased executive compensation, although not necessarily a full mediation effect, merely a partial one. This implies that there are other interpretive paths that can be explored in the future. In addition, prior research has shown that VC monitoring reduces the sensitivity between executive compensation and corporate performance [42,84], suggesting a substitute effect between VC monitoring and increased executive compensation. Therefore, it is possible that investigations regarding the influence of VC on executive compensation under different monitoring costs could be conducted in future research.

**Author Contributions:** Conceptualization, H.Z.; methodology, L.J.; software, L.J.; validation, H.Z. and L.J.; formal analysis, L.J.; investigation, L.J.; resources, L.J.; data curation, L.J.; writing—original draft preparation, L.J.; writing—review and editing, H.Z. and L.J.; visualization, L.J.; supervision, H.Z.; project administration, H.Z.; funding acquisition, H.Z. All authors have read and agreed to the published version of the manuscript.

**Funding:** This research received no external funding.

**Institutional Review Board Statement:** Not applicable.

**Informed Consent Statement:** Not applicable.

**Data Availability Statement:** The data presented in this study are available on request from the corresponding author.

**Conflicts of Interest:** The authors declare no conflict of interest.

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
