# Peer review of "Venture Capital, Compensation Incentive, and Corporate Sustainable Development"

_sustainability, doi:10.3390/su15075899_

Round 1
Reviewer 1 Report
Dear Authors,
Please find below and attached my comments and suggestions for your work.
Good luck!
Kind regards,
The Reviewer
Review Report Form
Journal: Sustainability (ISSN 2071-1050)
Manuscript ID: sustainability-2269285
Type: Article
Title: Venture capital, Compensation Incentive and Corporate Sustainable Development
Authors: Li Jing * , Huying Zhang
Section: Economic and Business Aspects of Sustainability
Special Issue: Insights on Venture Capital and Sustainable Development of Enterprise
Submission Date: 22 February 2023
Dear Authors,
I have carefully analyzed your article entitled “Venture capital, Compensation Incentive and Corporate Sustainable Development”.
Congratulations for your work and valuable insights reflected in the content of the manuscript!
The structure of my Review Report Form takes into consideration two sections, namely: (A.) General overview of the article and strong points; and (B) Suggestions meant to improve your current manuscript.
(A.) General overview of the article and strong points:
Ø General background of the study and aim of the study: The authors have mentioned that venture capital (VCs) can raise executive compensation and corporate innovation. What is more, the authors have prompted that previous studies have also indicated that compensation incentives can be beneficial to corporate innovation. Furthermore, the authors have stated that, even though the relationship between these three variables has been validated with any two of them, the relationship between VCs, executive compensation, and corporate innovation has not received enough consideration.
Ø Results of the study: In terms of the results of this current study, it ought to be mentioned that that the authors have mentioned that their research focused on the connection among these three variables, and we found that VCs have a mediating effect on innovation through executive compensation incentives, although not necessarily a full mediation effect, merely a partial one. Moreover, the authors stressed that they have found that VCs primarily play the role of compensation incentive by amplifying the internal salary gap of the corporate. In addition, we discovered that experienced VCs or companies with lower governance quality are more likely to use compensation incentive in order to promote corporate innovation. Also, the authors have brought to the attention that this study provides valuable insight for VCs in cultivating corporate innovation, as well as for corporate looking to boost their innovation.
(B) Suggestions meant to improve your current manuscript:
Distinguished Authors I would kindly like to suggest the following aspects:
(1.) Closely analyzing the article, since there are some English language improvements and slight corrections that need to be taken care of. Thus, my recommendation would be to carefully proofread the entire manuscript.
(2.) Also, I have closely analyzed the format of the article, in order to check whether it follows the guidelines which are specific to the publisher. Thus, I have noticed that the current form of your work needs improvement in this regard. So, my kind suggestion is to closely analyze again the guidelines belonging to the publisher, since the article should fit exactly the publisher’s guidelines. For instance, the keywords, the subsections, the references, currently do not fit the style and the requirements of the publisher. Also, it would be highly recommendable to include in the abstract of your study more highly relevant details that refer to the research objectives and the methodology used. This would definitely be considered a plus for your scientific work.
(3.) In continuation, the suggestion would also be inserting in your article a few ideas concerning the correlation between effects of the COVID-19 pandemic and the COVID-19 global crisis, sustainability, and Sustainable Development Goals, while focusing on the venture capital, compensation incentive and Corporate Sustainable Development, since these are key focuses these days. In this context, I had the chance to read a few interesting scientific works recently, among which I would like to mention: Corporate Social Responsibility, Corporate Governance and Business Performance: Limits and Challenges Imposed by the Implementation of Directive 2013/34/EU in Romania. Sustainability 2019, 11, 5146. https://doi.org/10.3390/su11195146; OECD. Measuring the Impacts of Business on Well-Being and Sustainability. https://www.oecd.org/statistics/Measuring-impacts-of-business-on-well-being.pdf; OECD. 2022. Toward sustainable economic development through promoting and enabling responsible business conduct. https://www.oecd-ilibrary.org/sites/f7813858-en/index.html?itemId=/content/component/f7813858-en.
Dear Authors, congratulations once again for your work and valuable insights reflected in the content of the manuscript, and I hope my comments will be of value to you!
Kind regards,
The Reviewer

Author Response
Dear Reviewers:
Thanks very much for taking your time to review this manuscript (sustainability-2269285). We are very grateful to Reviewer for reviewing the paper so carefully.We have carefully considered the suggestion of Reviewer and make some changes. Thanks again!
Comment 1:Closely analyzing the article, since there are some English language improvements and slight corrections that need to be taken care of. Thus, my recommendation would be to carefully proofread the entire manuscript.
Response 1: Above all, thanks very much for the comments, which are very helpful to improve the quality of this article. We have revised the manuscript and especially paid much attention to your comments and suggestions.
We regret there were problems with the English.The paper has been carefully revised by a professional language editing service to improve the grammar and readability. The paper has undergone English language editing by MDPI. The text has been checked for correct use of grammar and common technical terms, and edited to a level suitable for reporting research in a scholarly journal. MDPI uses experienced, native English speaking editors. Full details of the editing service can be found in the website (https://www.mdpi.com/authors/english)
Comment 2: Also, I have closely analyzed the format of the article, in order to check whether it follows the guidelines which are specific to the publisher. Thus, I have noticed that the current form of your work needs improvement in this regard. So, my kind suggestion is to closely analyze again the guidelines belonging to the publisher, since the article should fit exactly the publisher’s guidelines. For instance, the keywords, the subsections, the references, currently do not fit the style and the requirements of the publisher. Also, it would be highly recommendable to include in the abstract of your study more highly relevant details that refer to the research objectives and the methodology used. This would definitely be considered a plus for your scientific work.
Response 2:Thanks very much for your comments, which are very helpful to improve the quality of this article. We have done it according to your ideas.
We have checked it to follow the guidelines which are specific to the publisher, especially in terms of article structure, reference format, etc., we have adjusted them again.
We have rewrite the abstract to include more highly relevant details that refer to the research objectives and the methodology used. As following:
Venture capital (VC) can increase executive compensation and corporate innovation. Previous studies have also indicated that compensation incentives can be beneficial to corporate innovation. Even though the relationships between two of these three variables have been validated, the relationship between VC, executive compensation, and corporate innovation has not received enough consideration. Our research focused on the connections among these three variables, and we chose corporate as our sample, which listed corporations on the Shenzhen and Shanghai stock exchanges in the period from 2009 to 2017. We found that VC has a mediating effect on innovation through executive compensation incentives, although not necessarily a full mediation effect—merely a partial one. Moreover, we found that VC primarily plays the role of a compensation incentive by amplifying the internal salary gap of corporate. By employing invention patents to replace explanatory variables, using a Heckman two-stage method, and utilizing propensity score matching (PSM) for robustness testing, the validity of the conclusion was confirmed. In addition, we discovered that experienced VC or companies with lower governance quality are more likely to use compensation incentives to promote corporate innovation. This study provides valuable insight for VC in cultivating corporate innovation, as well as for corporates looking to boost their innovation.
Comment 3:In continuation, the suggestion would also be inserting in your article a few ideas concerning the correlation between effects of the COVID-19 pandemic and the COVID-19 global crisis, sustainability, and Sustainable Development Goals, while focusing on the venture capital, compensation incentive and Corporate Sustainable Development, since these are key focuses these days. In this context, I had the chance to read a few interesting scientific works recently, among which I would like to mention: Corporate Social Responsibility, Corporate Governance and Business Performance: Limits and Challenges Imposed by the Implementation of Directive 2013/34/EU in Romania. Sustainability 2019, 11, 5146. https://doi.org/10.3390/su11195146; OECD. Measuring the Impacts of Business on Well-Being and Sustainability. https://www.oecd.org/statistics/Measuring-impacts-of-business-on-well-being.pdf; OECD. 2022. Toward sustainable economic development through promoting and enabling responsible business conduct. https://www.oecd-ilibrary.org/sites/f7813858-en/index.html?itemId=/content/component/f7813858-en.
Response 3: Thank you for this valuable feedback.
In the Conclusions and Discussion part, We add the venture capital, compensation incentive and Corporate Sustainable Development, as following:
Against the backdrop of the COVID-19 pandemic, the impact of corporate governance and social responsibility regarding sustainable development has been widely scrutinized [77-79]. Our findings suggest that venture capital can improve the innovation performance of enterprises by participating in corporate governance through enhancing managerial compensation, thereby affecting the sustainable development of enterprises. However, given the crucial importance of corporate social responsibility for sustainable development, the impact of venture capital on corporate social responsibility and sustainable development deserves further attention as a potential area of future investigation.

Reviewer 2 Report
1. The article "Venture Capital, Compensation Incentive and Corporate Sustainable Development" explores the relationship between venture capital (VC), executive compensation, and corporate innovation.
2. The results show that venture capital can foster corporate innovation through compensation incentives, and that these types of incentives are mainly related to widening the internal wage gap within the corporation.
3. The results of the study confirmed the indirect effect of venture capital on corporate innovation, which means that the effect of venture capital on corporate innovation does not depend only on increasing executive salaries, and that analysis of the effects of other factors requires further research.
4. Statistical robustness tests showed that the findings remain valid.
5. The main comments relate to the fact that this paper uses the traditional assessment of the impact of venture capital on corporate innovation growth, relying solely on the number of patents, which does not always accurately measure the potential for corporate innovation, since corporate innovation may include corporate secrets or processes and technological innovations that cannot be accurately measured using only patent data, so it is advisable to rely on more adequate and appropriate indicators in further research
6. Otherwise, the work is of high quality.
Author Response
Dear Reviewers:
Thanks very much for taking your time to review this manuscript (sustainability-2269285). We are very grateful to Reviewer for reviewing the paper so carefully.We have carefully considered the suggestion of Reviewer and make some changes. Thanks again!
Comment 1: The main comments relate to the fact that this paper uses the traditional assessment of the impact of venture capital on corporate innovation growth, relying solely on the number of patents, which does not always accurately measure the potential for corporate innovation, since corporate innovation may include corporate secrets or processes and technological innovations that cannot be accurately measured using only patent data, so it is advisable to rely on more adequate and appropriate indicators in further research
Response 1: Above all, thanks very much for the comments, which are very helpful to improve the quality of this article. We have revised the manuscript and especially paid much attention to your comments and suggestions.
Your suggestions are extremely important, and we have also taken note of this issue. As you mentioned, innovation is not just about patent applications, but also includes secrets and other factors. Relying solely on patent applications to measure corporate innovation is not very accurate. However, despite the limitations of using patents to measure corporate innovation, most studies still use patents as a measure due to the difficulty of measuring trade secrets and other factors. In addition, using the number of patents is relatively more objective. Additionally, we also emphasized in the limitations section of the article that using patents to measure corporate innovation is not entirely accurate, and that new and more accurate indicators can be developed in the future to measure corporate innovation.
